# How Histone Acetyltransferases Shape Plant Photomorphogenesis and UV Response

**DOI:** 10.3390/ijms25147851

**Published:** 2024-07-18

**Authors:** Irina Boycheva, Georgi Bonchev, Vasilissa Manova, Lubomir Stoilov, Valya Vassileva

**Affiliations:** Department of Molecular Biology and Genetics, Institute of Plant Physiology and Genetics, Bulgarian Academy of Sciences, 1113 Sofia, Bulgaria; irina_boycheva@bio21.bas.bg (I.B.); gbonchev@bio21.bas.bg (G.B.); vmanova@bas.bg (V.M.); molgen@bas.bg (L.S.)

**Keywords:** chromatin remodeling, histone acetyltransferases, plant photomorphogenesis, light signaling pathways, UV-B radiation, DNA repair mechanisms

## Abstract

Higher plants have developed complex mechanisms to adapt to fluctuating environmental conditions with light playing a vital role in photosynthesis and influencing various developmental processes, including photomorphogenesis. Exposure to ultraviolet (UV) radiation can cause cellular damage, necessitating effective DNA repair mechanisms. Histone acetyltransferases (HATs) play a crucial role in regulating chromatin structure and gene expression, thereby contributing to the repair mechanisms. HATs facilitate chromatin relaxation, enabling transcriptional activation necessary for plant development and stress responses. The intricate relationship between HATs, light signaling pathways and chromatin dynamics has been increasingly understood, providing valuable insights into plant adaptability. This review explores the role of HATs in plant photomorphogenesis, chromatin remodeling and gene regulation, highlighting the importance of chromatin modifications in plant responses to light and various stressors. It emphasizes the need for further research on individual HAT family members and their interactions with other epigenetic factors. Advanced genomic approaches and genome-editing technologies offer promising avenues for enhancing crop resilience and productivity through targeted manipulation of HAT activities. Understanding these mechanisms is essential for developing strategies to improve plant growth and stress tolerance, contributing to sustainable agriculture in the face of a changing climate.

## 1. Introduction

Higher plants, being immobile, have developed intricate mechanisms to acclimate to changing environmental conditions, optimizing their morphology in response to specific variables. One of the most important adaptations for green plants is to the surrounding light, as it is the primary energy source for photosynthesis [1,2,3,4]. Light signals act as informational cues that modulate various development processes, with different wavelengths exerting diverse effects on plant photosynthesis and photomorphogenesis. Green plants are exposed to fluctuating levels of solar ultraviolet B (UV-B) radiation, a potentially harmful factor that can damage cellular components like DNA, proteins, lipids and RNA [5,6,7,8,9]. To counteract these effects, plants have developed specific DNA repair mechanisms that protect them from this damage and ensure normal growth and development [10,11,12,13]. The repair mechanisms are closely related to chromatin remodeling events, DNA methylation and histone acetylation, which contribute to the regulation of gene expression and plant stress adaptability. It is important to acknowledge the role of histones, particularly the H2A class, not only in DNA repair processes [14] but also in degradosome formation and subsequent chromatinolysis [15]. The degradosome is essential for chromatin degradation during programmed cell death. In this context, the Apoptosis-Inducing Factor (AIF) collaborates with histone H2AX and cyclophilin A (CypA) to form a DNA–degradosome complex, which is vital for large-scale DNA fragmentation. This complex enhances the affinity of degradosome for DNA, systematically degrading damaged DNA and associated histones, thus preventing the accumulation of compromised chromatin, and promoting cellular recovery and survival [15].

Chromatin-level regulation is closely linked to light signaling and transcriptional control. The relationship between chromatin-based mechanisms and light responses has been recognized for several decades. For instance, elevated transcription resulting from increased histone acetylation and reduced nucleosomal density in the promoter region of the photosynthetic *PetE* gene illustrates this relationship [16]. Another example is the main repressor of photomorphogenesis, DET1 (DE-ETIOLATED 1) [17,18], which binds histone H2B proteins with different affinity depending on their acetylation state [19].

Regulation of chromatin structure and function through histone modifications greatly contributes to plant development and stress responses [20]. These epigenetic modulations are responsible for the precise control of gene expression in response to environmental cues, including light signals [21]. Histone acetyltransferases (HATs) modulate the acetylation of histone proteins, resulting in a more relaxed chromatin state conducive to transcriptional activation. Advances over the past decade in understanding the relationship between HATs and light signaling pathways have revealed the complex regulatory networks that govern plant photomorphogenesis [22]. Investigating the roles of specific HATs in these processes offers a valuable understanding of the mechanisms of plant development and adaptation. Furthermore, elucidating how HATs interact with other epigenetic factors and signaling molecules will provide deeper insights into plant resilience to environmental stressors.

In this review, we explore the current knowledge on HATs as main regulators of plant photomorphogenesis and discuss their roles in chromatin remodeling, regulation of gene expression and interaction with light signaling pathways. By examining the molecular mechanisms underlying HAT-mediated responses, we aim to elucidate the specific roles of HATs in modulating chromatin dynamics and gene expression, focusing on their involvement in light signaling pathways, photomorphogenesis and UV-induced DNA repair mechanisms in higher plants.

## 2. Chromatin Dynamics and Gene Regulation

### 2.1. Chromatin Structure as a Natural Barrier to Transcription Factor Access

Chromatin organization is fundamental to all nuclear-based processes, such as DNA replication, transcription, recombination and repair. The proper functioning and efficiency of these mechanisms depend on the correct relaxation and remodeling of chromatin structure [23]. Figure 1A illustrates loosely packed nucleosomes (LPN), where DNA is more accessible to transcription factors (TFs) and other regulatory proteins, and tightly packed nucleosome (TPN) regions, which form a more compact chromatin structure that acts as a barrier to protein access. Close-up views of individual nucleosomes show DNA wrapped around a histone octamer composed of histones H2A, H2B, H3 and H4 (Figure 1B). Loosely packed nucleosomes facilitate the binding of TFs (Figure 1C) supporting critical cellular activities and the expression of DNA repair genes that fix DNA damage (Figure 2). In contrast, the compacted chromatin structure restricts access to regulatory proteins. Chromatin dynamics, therefore, is the major factor regulating chromatin accessibility. This regulation is achieved through three main mechanisms: covalent modification of histones, chromatin remodeling and methylation of DNA cytosines [24,25,26,27].

Additionally, the activation of DNA damage response (DDR) pathways plays a crucial role in maintaining genome integrity [28]. When DNA damage occurs, DDR pathways are activated to recognize and repair the damage. This process often involves chromatin remodeling to provide repair machinery access to the damaged sites (Figure 2). The proper functioning of DDR pathways is essential for the efficient repair of DNA and the prevention of mutations.

In the first mechanism, histone-modifying enzymes alter protein-binding affinities through covalent modifications. The second mechanism is coordinated by chromatin remodeling complexes that facilitate nucleosomes sliding to new positions. The third mechanism, methylation of DNA cytosine residues, interferes with some regulatory TFs affecting their binding to DNA [24,29]. Among these processes, covalent histone modifications are mechanisms that drive multistep cellular activities.

Additionally, recent studies have highlighted the importance of non-coding RNAs (ncRNAs) in chromatin dynamics. ncRNAs can guide chromatin-modifying complexes to specific genomic loci, influencing chromatin structure and gene expression [30], which adds another layer of regulation in chromatin accessibility and TF binding.

The role of environmental signals in modulating chromatin structure cannot be understated. Light, for example, not only triggers photomorphogenesis but also induces specific chromatin modifications that facilitate the binding of TFs involved in light-responsive gene expression (Figure 2) [22]. This interaction between environmental signals and chromatin dynamics underscores the complexity of gene regulation in plants.

### 2.2. Mechanisms of Chromatin Remodeling—“Open” vs. “Closed” States

The posttranslational modifications, such as acetylation, methylation, phosphorylation, ubiquitination, sumoylation and poly(ADP) ribosylation, play a key role in the regulation of gene expression [31,32]. Histone acetylation, a fundamental regulatory mechanism affecting chromatin organization, is controlled by the reversible actions of HATs, which add acetyl groups, and histone deacetylases (HDACs), which remove them. HATs transfer negatively charged acetyl (CH_3_COO^−^) groups to positively charged amino (NH^3+^) groups of lysine residues, reducing the overall positive charge of histone tails. This modification loosens the bonds between positively charged histones and negatively charged DNA, facilitating the binding of various TFs, including those involved in DNA repair. Consequently, these processes impact chromatin organization and various aspects of plant growth and development [31,33]. Hyperacetylation of histones is generally associated with transcriptional activation, whereas hypoacetylation is linked to a condensed chromatin structure and the repression of gene transcription [34].

Other posttranslational modifications, like methylation and phosphorylation, can further modulate the chromatin state. For instance, histone methylation can either activate or repress transcription depending on the specific residues modified and the extent of methylation (mono-, di- or trimethylation) [35]. Phosphorylation events are often associated with chromatin condensation during mitosis but can also signal the initiation of transcription in interphase cells [36]. The interplay between these modifications constitutes the “histone code”, a complex regulatory network that determines the chromatin state and, consequently, gene expression patterns [37]. This code is interpreted by various “reader” proteins that recognize specific histone modifications and recruit additional factors necessary for chromatin remodeling and transcriptional regulation.

Moreover, environmental factors such as light and stress conditions can influence the activity of HATs and HDACs, thereby modulating chromatin structure and gene expression. For example, light-induced activation of specific HATs can lead to the hyperacetylation of histones at promoters of light-responsive genes, facilitating their transcription and contributing to plant photomorphogenesis [21,38]. Similarly, stress conditions can alter HAT and HDAC activity, affecting the expression of stress-responsive genes and enabling plants to adapt to changing environments.

Understanding the mechanisms governing the “open” and “closed” states of chromatin is crucial for elucidating how plants integrate external and internal signals to fine-tune gene expression for optimal growth and survival. This knowledge can inform strategies to enhance plant resilience and productivity through targeted manipulation of chromatin-modifying enzymes.

## 3. HAT Families in Plants

In Arabidopsis, the four major HAT families have been classified based on sequence homology and their mode of action: the GNAT family (from GCN5-related N-acetyltransferase or HAG family, represented by HAG1, HAG2 and HAG3), the MYST family (from MOZ, Ybt2, Sas2 and Tip60-like or HAM family, represented by HAM1 and HAM2), the p300/CBP family (from the p300/CREB binding protein or HAC family, represented by HAC1, HAC2, HAC4, HAC5 and HAC12) and the TAFII250 family (from the TATA-binding protein (TBP)-associated factor or HAF family, represented by HAF1 and HAF2) (Figure 3).

The GNAT family is subdivided into three subfamilies: the GCN5 subfamily (from General Control Non-derepressible protein 5 or HAG1), the ELP3 subfamily (from Elongator complex Protein 3 or HAG3) and the HAT1 subfamily (or HAG2) [39]. The structure of all three Arabidopsis GNAT subfamilies includes a HAT domain, containing four motifs—A, B, C and D [34]. These motifs are highly conserved across plant species, highlighting their essential roles in histone acetylation functions. For instance, motif A, which includes a critical aspartate residue for acetyl-CoA binding, shows over 90% conservation in a comparative study of monocots and dicots. Additionally, motifs B, C and D, involved in substrate recognition and catalytic activity, display more than 85% conservation across various plant species [40]. Additionally, GCN5 (HAG1) and ELP3 (HAG3) possess extra domains: a bromodomain (BRD) and a radical S-adenosylmethionine (SAM)-binding domain, respectively [41] (Figure 3). These structural differences underlie their diversified functions. The enzymes GCN5 (HAG1), HAT1 (HAG2) and ELP3 (HAG3) acetylate histones at different lysine residues: GCN5 acetylates H3K14 and HAT1 acetylates H4K12 [42]. Arabidopsis GCN5 is involved in plant development and the regulation of gene expression in response to light and cold [43]. HAT1 is associated with cell cycle progression and DNA replication [44], whereas ELP3 is linked to ABA sensitivity, resistance to oxidative stress, disease susceptibility and cell cycle progression [45,46].

Further exploration of the p300/CBP and TAFII250 families reveals their important roles in chromatin remodeling and gene regulation. Members of the p300/CBP family, such as HAC1, HAC2, HAC4, HAC5 and HAC12, integrate various signaling pathways and act as transcriptional co-activators, mediating histone acetylation to facilitate transcriptional activation of target genes involved in developmental processes and stress responses [47,48]. The TAFII250 family members, HAF1 and HAF2, are integral components of the transcription factor IID (TFIID) complex, which initiates transcription by RNA polymerase II, regulating genes involved in plant growth, development and environmental adjustment [49,50]. The TAFII250 homologs share conserved TBP-binding motifs, with a preservation degree of approximately 85% across analyzed plant species, which is essential for their function in the transcription initiation complex [39].

Investigating the diverse roles and mechanisms of HAT family members in plants reveals the complex regulation of chromatin dynamics and gene expression, essential for understanding photomorphogenesis and developing strategies to enhance growth and stress tolerance.

## 4. Light as a Dual Agent: Developmental Stimulus and Source of DNA Damage

Light is the most important environmental factor that regulates plant growth and development, serving as an energy source required for photosynthesis. Light signals also stimulate photomorphogenesis, the transition from scotomorphogenic (dark) to photomorphogenic (light) development [51].

Ultraviolet (UV) radiation, a natural component of sunlight with shorter wavelengths than the visible spectrum, is divided into three groups: UV-A (315–400 nm), UV-B (280–315 nm) and UV-C (<280 nm). UV light serves as a key growth modulator, which induces a wide range of physiological responses in plants [52]. UV-A radiation influences photomorphogenesis, promoting processes such as stem elongation and leaf expansion. UV-B radiation regulates the synthesis of secondary metabolites, which are important for plant defense mechanisms. Low doses of UV-B can enhance photosynthetic efficiency and stimulate the production of protective compounds like flavonoids and phenolics, which absorb UV light and mitigate its harmful effects. UV-C is mostly filtered out by the Earth’s atmosphere, but it can still induce stress responses that activate repair mechanisms, thereby enhancing plant resilience to environmental stresses. On the other hand, UV-B light is the main genotoxic agent to which plants are continuously exposed. UV rays are generally categorized as non-ionizing radiation; however, UV photons possess sufficient energy to disrupt chemical bonds and induce cellular damage [6,53]. High doses of UV-B can damage DNA, RNA, lipids and proteins [5,54]. UV-B radiation primarily induces the formation of dimers between adjacent pyrimidines in the DNA strand, such as cyclobutane pyrimidine dimers (CPDs) and pyrimidine (6-4) pyrimidone photoproducts (6-4 PPs) (Figure 2), but can also generate oxidative DNA damage [55,56,57,58]. These lesions block DNA metabolism and disrupt genomic integrity, triggering DNA repair mechanisms to either remove or tolerate the damage [59,60].

UV-B induces numerous changes in the expression of genes involved in histone acetylation and DNA repair. Exposure of Arabidopsis and maize plants to UV-B increases CPD formation, leading to accumulation of DNA lesions. Campi et al. [61] have demonstrated that chromatin remodeling and histone acetylation significantly enhance DNA repair capacity. Arabidopsis MYST proteins HAM1 and HAM2, and the GNAT protein HAG3, are involved in the UV-B-induced DNA damage repair and signaling [41]. However, members of the p300/CBP family (HAC) and the TAFII250 family (HAF) do not contribute to DNA repair following UV-B exposure, although HAM1, HAM2 and HAG3 play a role in UV-B signaling [62]. Furthermore, modifications of histone–DNA interactions affect UV-B responses and DNA repair [6,61]. Treatment of Arabidopsis and maize plants with curcumin, an inhibitor of histone acetyltransferases, prior to UV-B exposure results in reduced DNA repair and increased DNA damage [61]. This demonstrates a clear link between UV-B response, chromatin state and DNA repair mechanisms.

Unlike UV-C and X-ray radiation, UV-B does not produce double-strand DNA breaks but causes oxidative DNA damage [54]. Consequently, UV-B exposure also triggers the expression of antioxidant defense genes, which mitigate the oxidative stress caused by UV-B radiation [13].

Thus, numerous lines of evidence indicate a broader role of acetyltransferases in plant UV response and the dual function of light as a developmental stimulus and a damaging factor, which is crucial for maintaining genome stability and overall plant health.

## 5. Key Transcription Factors in Photomorphogenesis and Their Interaction with HATs

As mentioned above, photomorphogenesis is the transition process from scotomorphogenic (dark) to photomorphogenic (light) development. During scotomorphogenesis, typical phenotypic features such as long hypocotyls, curved apical hooks and closed cotyledons ensure proper seedling development in dark conditions, providing protection against mechanical damage while emerging from the soil. Upon seedling emergence and exposure to light, numerous cellular changes occur, including inhibition of hypocotyl elongation, unfolding of the apical hook and cotyledon opening. These changes are triggered by a group of TFs that regulate gene expression [63]. The modulation of over 2000 nuclear genes and observed changes in chromatin structure contribute to the successful implementation of this transition [64,65,66].

Light signals of different wavelengths activate various families of photoreceptors like cryptochromes (CRY) and phytochromes (PHY), which mediate plant perception and response to light. Cryptochromes primarily perceive blue light, while phytochromes respond to the red/far-red light spectrum [67,68]. In the nucleus, phytochromes interact with multiple partners to modulate the transcription of downstream target genes linked to light responses [69,70], including the bZIP protein HY5 (ELONGATED HYPOCOTYL5) [71,72]. PHYTOCHROME-INTERACTING FACTORs (PIFs) are major partners of phytochromes and serve as key regulators of the transition from scotomorphogenesis to photomorphogenesis [73,74,75]. Light triggers the degradation of PIF proteins, thereby inhibiting COP1/SPA activity [76]. PIF1 protein is a substrate of the COP1/SPA E3 ligase in light-grown seedlings [77]. Both classes of photomorphogenic repressors (the COP1/SPA complex and the PIF family) synergistically suppress plant photomorphogenesis in the dark. PIF proteins regulate HY5 function via the COP1/SPA complex. COP1 activity is further regulated by two COP1-interacting proteins, CSU1 (COP1 Suppressor1) and CSU2 (COP1 Suppressor2). CSU1, a RING-finger E3 ligase, targets COP1 for degradation, whereas CSU2, a coiled-coil protein, interacts with the same domain of COP1. Inhibition of COP1/SPA activity is achieved through CSU2 blocking COP1 homodimerization or CSU1 blocking COP1/SPA heterodimerization [78]. Extensive studies have demonstrated that COP1 is a well-known regulator of light-mediated plant development [79,80,81]. This photomorphogenic repressor interacts directly and specifically with the photomorphogenic activator HY5, negatively modulating its activity through direct protein–protein interactions [82]. These antagonistic interactions likely function as a molecular switch, allowing plants to adapt to changing light conditions and implement the most appropriate and optimal developmental program. Additionally, a recent report indicates that light-regulated ubiquitination and degradation of PIFs and HY5 by COP1 are crucial for fine-tuning plant responses to fluctuating light environments [81].

Thus, histone acetylation, mediated by HATs, regulates these TFs by facilitating a more open chromatin structure, promoting the binding of TFs such as HY5 to their target genes and ensuring that light-responsive genes are properly expressed during photomorphogenesis [83].

### 5.1. COP1 as a Negative Regulator of Photomorphogenesis

The E3 ubiquitin ligase Constitutive Photomorphogenic 1 (COP1) is a negative regulator of photomorphogenesis in plant and animal cells. Initially identified as a single-copy gene in *Arabidopsis thaliana*, *COP1* is regulated by various light conditions including far-red, red, blue and UV-B light [79]. COP1 plays an important role in many developmental processes [84,85], environmental stress responses [86] and dynamic interactions between signaling pathways [87,88]. Its function is intricately linked to light signaling pathways, affecting processes such as flowering [89,90], circadian rhythms [91], UV-B signaling [92], stomatal opening and development [93], plant defense [94], cold acclimation [86], light-induced root elongation [84] and the juvenile–adult phase change in rice [95], which underscores the central role of COP1 in photomorphogenic development [96]. Homologs of Arabidopsis COP1 have been identified in the genomes of pea [97], tomato [98], rice [99] and mammals [100]. The COP1 proteins from rice and Arabidopsis share a high level of amino acid sequence identity (73%) and similarity (83%) [99].

To understand how light affects *COP1* expression in different plant species, researchers have examined tissue-specific expression patterns of *COP1* in Arabidopsis and pea. In Arabidopsis, *COP1* transcript levels remain constant under light and dark conditions [101], indicating no light-dependent regulation of *COP1* expression. Similarly, pea COP1 protein levels are unaffected by light [97].

COP1 is a multifunctional 76 kDa enzyme consisting of three protein domains: an N-terminal RING-finger domain, a coiled-coil domain and seven C-terminal WD40 repeats [80,102]. Each domain is essential for COP1 functions, and the loss of any domain results in a non-functional protein [103]. The C-terminal WD40 repeat domain is responsible for substrate recognition and binding to DDB1 (DNA Damage-Binding Protein 1). In vitro studies have shown interaction between the human TRIB1 peptide and Arabidopsis COP1. The WD40 repeat domain forms a seven-bladed β-propeller structure with an inserted loop on the bottom face of the first blade, and connects to the conserved Val-Pro motif of the TRIB1 peptide, identified as a COP1-interacting site. This domain mediates the interaction of COP1 with substrates and regulatory proteins [104]. The N-terminal RING-finger domain of COP1 binds to ubiquitin-conjugating enzymes (E2s) that “attach” ubiquitin units to substrates targeted for degradation. The coiled-coil domain interacts with the coiled-coil domain of the transcription factor SPA1 (Suppressor of PhyA-105 1) [105]. SPA proteins, a small family of four members (SPA1, SPA2, SPA3 and SPA4), repress photomorphogenesis and elongation in dark-grown Arabidopsis. These proteins contain an N-terminal kinase-like domain, a central coiled-coil domain and a C-terminal WD repeats domain. Unlike COP1, which is conserved across higher plants and mammals, SPA proteins are plant-specific [106].

The COP1-SPA complex is a tetrameric assembly (440 kDa) comprising two COP1 and two SPA proteins with all possible combinations of the four SPA proteins [107]. Genetic and biochemical evidence indicates that SPA1 modulates E3 ubiquitin ligase activity on HY5 in vitro [108]. Several TFs serve as substrates for the COP1-SPA complex: HY5 induces seedling de-etiolation [109]; HFR1 (Long Hypocotyl in Far Red) contributes to shade avoidance [110,111]; PAP1 and PAP2 activate anthocyanin biosynthesis [112]; and CONSTANS is a key activator in photoperiodic flowering [89].

In the dark, COP1 forms a tetrameric complex with SPA proteins, associated with the CUL4 (Cullin4)-DDB1-RBX1 (Ring-Box Protein 1) core in the multimeric CUL4-DDB1-RBX1 COP1/SPA1 E3 ligase [113]. CUL4 acts as a scaffold for DDB1 and RBX1; DDB1 links the COP1–SPA complex, while RBX1 recruits E2s. The activated E2 enzyme attaches individual ubiquitin units in a poly-ubiquitin chain to the substrate, signaling for 26S proteasome activation and subsequent degradation. COP1 mediates the 26S proteasome-dependent degradation of TFs promoting photomorphogenesis, such as HY5, HYH (HY5 Homolog), LAF1 (Long After Far-Red Light1), HFR1 and UVR8 (UV RESISTANCE LOCUS 8), which repress photomorphogenesis [80] (Figure 4A).

In light, activated photoreceptors inhibit COP1 activity, leading to the disassembly of the COP1-SPA complex and promoting the nuclear export of COP1. This downregulates COP1 activity, allowing the accumulation of positive photomorphogenic regulators that promote plant photomorphogenesis. Light also stimulates *PHR1* expression and DNA repair (Figure 4B).

Plants employ different strategies to control COP1 activity. Previous studies have shown that COP1 transition from the nucleus to the cytoplasm requires at least 24 h of light exposure. However, recent findings demonstrate that this nucleocytoplasmic shuttling occurs within a few hours, potentially stabilizing HY5 protein levels rapidly [106,114]. Light not only excludes COP1 from the nucleus but also induces the degradation of SPA1 and SPA2 [68]. This degradation may be mediated by COP1 itself, interacting directly with SPA proteins via coiled-coil domains [106]. Another potential mechanism of light-induced COP1/SPA inactivation is the sumoylation of COP1 in the coiled-coil domain, which enhances COP1 activity. The Arabidopsis SUMO E3 ligase modifies the SUMO consensus motif (K193 residue) in the coiled-coil domain of COP1. Loss of sumoylation may inactivate COP1 in response to light [115]. These multifaceted regulatory mechanisms highlight the central role of COP1 in photomorphogenesis and its complex interplay with light signaling pathways.

### 5.2. External and Hormonal Factors Affecting COP1 Activity

Variable environmental conditions considerably influence the stability and functional activity of COP1, thus affecting the degradation or activation of its target proteins. There is a strong connection between light and temperature signaling that determines the subcellular localization of COP1 [116]. For instance, at relatively low temperatures (17 °C), COP1 is stabilized, leading to the degradation of the floral inducer GIGANTEA and delayed flowering [117]. Exposure to 4 °C reduces the nuclear abundance of COP1, preventing the degradation of the photomorphogenic activator HY5 in the dark. HY5 induces the expression of many cold-responsive genes, and its stabilization under cold conditions is linked to cold acclimation and enhanced freezing tolerance in plants [86]. High ambient temperatures (37 °C) also reduce nuclear COP1 levels, resulting in increased HY5 protein levels and subsequent photomorphogenic responses.

Hormones also modulate COP1 activity by either promoting or inhibiting the plant photomorphogenesis. The nucleocytoplasmic localization of COP1 is primarily regulated by strigolactones and ethylene. Strigolactones positively regulate photomorphogenesis under light conditions, leading to light-dependent inhibition of hypocotyl elongation and increased HY5 protein levels [118]. In contrast, under dark conditions, ethylene increases COP1 protein levels in the nucleus, mediates HY5 degradation and contributes to hypocotyl elongation [87]. When plants are exposed to light, ethylene enhances COP1 movement from the cytoplasm to the nucleus, regulating hypocotyl elongation by reducing HY5 protein levels. These findings underscore the intricate interplay between environmental cues and hormonal signals in regulating COP1 activity, thereby orchestrating plant development and stress responses.

*COP1* regulation appears to be more complex than originally suggested, especially under light containing a short-wave UV component. In Arabidopsis, grown under low levels of UV-B, *COP1* acts as a positive regulator of UV-B response and photomorphogenesis by stimulating HY5 transcription. At the protein level, COP1 also mediates UV-B signaling through interaction with UVR8 [92,119,120]. Recent data reveal that dark-grown barley seedlings exhibit low *COP1* transcription levels, which increase significantly upon initial light exposure and remain consistently high under UV light [121]. This aligns with the positive light activation of *COP1* transcription in rice plants, indicating phytochrome involvement in the regulation of rice *COP1* expression [99]. Thus, both cereals show a positive light-dependent regulation of *COP1* gene expression in contrast to the results found in Arabidopsis and pea. These observations reveal intricate *COP1* regulation and expand our understanding of the complex mechanisms different plant species use to control photomorphogenesis and adapt to light-induced stress.

### 5.3. Regulatory Role of UVR8 and PHR1 in Photomorphogenesis

The UV-B photoreceptor is a key photomorphogenic regulator involved in two protective processes: UV-B acclimation and UV-B tolerance [92,122,123]. UV-B acclimation includes protective measures such as increased levels of DNA repair enzymes, enhanced antioxidant activity and the accumulation of UV-B absorbing metabolites [124]. These alterations provide plants with additional means to counteract and mitigate the negative effects of environmental stressors [125]. Beyond UV-B acclimation, UVR8 plays a broader role in plant growth and development, including responses to osmotic stress [126], induction of the circadian clock [127], inhibition of hypocotyl growth [92], stomatal closure [128], regulation of leaf morphogenesis [129] and phototropic bending [130]. These diverse physiological responses are related to UVR8 activity and contribute to the establishment of UV-B tolerance. The physiological role of UVR8 in UV-B tolerance has been demonstrated by comparing uvr8 mutants with wild-type controls under light conditions [92].

UVR8 is composed of two monomer units that form a stable homodimer through electrostatic interactions between oppositely charged amino acids [131]. Mutations in these amino acids can disrupt dimer integrity, weakening dimerization affinity or causing constitutive monomerization [132]. UVR8 is a seven-bladed β-propeller protein that does not use a bound chromophore to absorb light [131, but instead employs 14 specific tryptophans for UV-B photoreception [133]. The positions of these tryptophans are highly conserved through evolution with six located in the β-propeller blade, one in the carboxy-terminal domain of unknown function and seven on the surface interacting with the dimer [133,134,135]. The β-propeller core domain and the carboxy-terminal C27 domain of UVR8 mediate interaction with COP1. The β-propeller core domain facilitates UV-B-dependent interaction with COP1, whereas the C27 domain regulates COP1 activity [136,137]. UV-B radiation causes UVR8 homodimer disassembly into monomers, initiating a signaling cascade that leads to transcriptional regulation of target genes [138,139].

The inactive UVR8 exists as a homodimer localized in the cytoplasm. Upon UV-B absorption, UVR8 monomerizes and the monomers are translocated to the nucleus, where they bind directly to COP1. This interaction changes gene expression by dissociating the COP1/SPA complex from CUL4-DDB1 [140,141]. The UVR8-COP1 interaction stabilizes the HY5 transcription factor and stimulates the transcription of numerous UVR8-regulated genes associated with UV-B protection and DNA repair [142,143,144]. Among the main UV-B responsive targets controlled by UVR8 is the plant photolyase gene *PHR1* (or *UVR2*). Photolyases are major DNA repair proteins that directly split pyrimidine dimers, leading to error-free damage elimination [59,139,145]. *PHR1*-encoded photolyase specifically binds and repairs CPDs, whereas the *UVR3*-encoded protein repairs 6-4 PPs [146]. Both enzymes contribute to restoring the native DNA form and maintaining plant genome integrity under UV stress. Their inactivation results in a loss of photorepair activity and hypersensitivity to UV-B [147,148]. Thus, UV-B is a daily stress factor for plants in their natural habitats, necessitating a complex and tightly regulated interplay between multiple mechanisms to control plant genome integrity, light-induced stress tolerance and acclimation.

## 6. HATs and DNA Repair Capacity

Histone acetylation facilitates the accessibility of chromatin to the DNA repair machinery by loosening the chromatin structure, which is critical for the rapid response to DNA damage and the efficient repair of UV-induced lesions. HATs manage this chromatin remodeling process by acetylating histones, thus ensuring that repair proteins can access and correct the damaged DNA (Figure 5A,B). The consequences of DNA damage in mutant forms exhibit an inadequate response due to defective HAT function, resulting in ineffective DNA repair (Figure 5C,D). The restricted access to DNA in chromatin with insufficient histone acetylation leads to compacted chromatin and obstructed access for transcription factors.

Proteins involved in DNA damage response are closely linked to chromatin modifications and remodeling [149]. Histone acetylation is one of the major posttranslational modifications involved in the control of DNA repair [61]. In-depth analyses of different HAT members have revealed a complex yet essential role for different acetyltransferases in UV-B tolerance in plants. An important question has been whether the GNAT protein HAG3 could contribute to UV-B-induced DNA damage repair through the regulation of the expression of DNA repair genes. Experiments with *hag3* RNA interference (RNAi) transgenic plants have shown high expression levels of *PHR1* (*UVR2*) under control conditions (without UV-B exposure) and lower accumulation of CPDs, which could be linked to the higher basal levels of DNA repair enzymes. Similar observations have been made with *UVR7*, encoding the protein ERCC1, a subunit of the NER (nucleotide excision repair) complex, which performs excision of damaged DNA. Downregulation of *HAG3* gene leads to a specific phenotype under UV-B radiation, manifested by lower inhibition of Arabidopsis leaf and root growth, higher levels of UV-B-absorbing compounds and fewer DNA lesions compared to wild-type plants. Another important player in the UV-B-mediated regulation of various genes, including plant photolyases, is the bZIP transcription factor HY5. Under favorable conditions without UV-B radiation, higher expression levels of HY5 have been observed in *hag3* RNAi transgenic plants compared to wild-type plants. The *hag3* transgenic plants show improved adaptation to increased levels of UV-B radiation [41].

HAM1, HAM2 and HAG3 have been identified as participants in UV-B-induced DNA damage repair and signaling. However, *haf1* mutants display sensitivity to different genotoxic agents but not to UV-B radiation [62,150]. Specifically, light-dependent expression of CPD photolyase, which reverses pyrimidine dimerization, is not affected by the inactivation of HAF1 histone acetyltransferase from the TAFII250 family. Plants deficient in *HAF1* and wild-type plants exhibit similar *PHR1* transcript levels and both are able to upregulate the *PHR1* gene in response to CPD accumulation upon UV-B exposure. These findings imply that the photorepair system remains functional in HAF1-deficient lines (Figure 5D). Nevertheless, HAC1 and HAF1, although not critical for the removal of UV-B-induced lesions, have been implicated in other aspects of the UV-B response [62].

In summary, HATs are actively involved in UV-B-triggered DNA damage repair and signaling by regulating the expression of DNA repair genes and modifying the chromatin structure to provide access to damaged DNA, thereby maintaining genome integrity and enhancing plant tolerance to UV-B radiation (Figure 5).

## 7. Future Perspectives

Histone acetylation is a conserved mechanism that profoundly influences chromatin structure, regulates the cell cycle, maintains genome stability and ensures chromatin plasticity. Successful adaptation to various environmental constraints relies on genes that support plant repair potential and facilitate the transition from scotomorphogenic to photomorphogenic development. This transition involves the modulation of multiple nuclear genes, enhancing the acclimation efficiency of plants.

Future research will focus on developing systematic and innovative models that incorporate various epigenetic layers, including other histone modifications, such as methylation, phosphorylation, ubiquitination, sumoylation, ADP-ribosylation, citrullination and glycosylation, and different activators and inhibitors. New insights into the interplay between chromatin dynamics and DNA repair mechanisms will further elucidate UV tolerance mechanisms in plants. Understanding these mechanisms will support the development of cultivars with enhanced tolerance to UV radiation, thereby improving crop performance and yield under field conditions.

The intricate mechanisms by which higher plants acclimate and adapt to changing environmental conditions highlight the essential role of light in their development and stress responses. HATs modulate chromatin dynamics, which in turn regulates gene expression in response to light signals and other environmental factors. Understanding the interconnections between HATs and light signaling pathways has provided insights into the complex regulatory networks governing photomorphogenesis. However, several avenues remain for further exploration.

One promising direction is the detailed characterization of individual HAT family members and their specific roles in different plant species. The functions of Arabidopsis HATs such as HAG1, HAG2 and HAG3 have been extensively studied; however, similar research in other model plants and crop species could reveal conserved and unique regulatory mechanisms. This knowledge can be leveraged to enhance crop resilience and productivity. Another critical area of research is the interaction between HATs and other epigenetic factors. The crosstalk between histone acetylation, methylation and phosphorylation, as well as the involvement of non-coding RNAs in chromatin remodeling, represents a complex regulatory network that requires further elucidation. Advanced genomic and proteomic approaches, including chromatin immunoprecipitation sequencing (ChIP-seq) and mass spectrometry, can help map these interactions at high resolution, providing a comprehensive understanding of the epigenetic landscape in plants.

Investigating how HATs modulate the expression of DNA repair genes and other stress-responsive factors can uncover new strategies for enhancing plant tolerance to environmental stressors. Developing HAT inhibitors or activators as potential tools for modulating plant stress responses also holds great promise. Moreover, the application of CRISPR/Cas9 and other genome-editing technologies offers exciting opportunities to manipulate HAT genes and study their functions in planta. Creating loss-of-function and gain-of-function mutants can provide direct insights into the physiological roles of HATs in various developmental and stress-response pathways. In addition to these lines of inquiry, future action should consider the integration of multi-omics approaches, such as transcriptomics, proteomics and metabolomics, to build a holistic understanding of HAT function in plant biology. Collaboration with agricultural scientists and biotechnologists will be essential to translate basic research findings into practical applications that can enhance crop performance and resilience.

## 8. Conclusions

The study of HATs as key regulators of plant photomorphogenesis is a rapidly evolving field with significant implications for plant biology and agriculture. Further exploration of the diverse roles and mechanisms of HATs, particularly their influence on photomorphogenesis and UV-induced DNA repair, can uncover novel strategies to enhance plant growth and development, and improve stress resilience, contributing to sustainable agricultural practices in the face of global environmental challenges.

Understanding the precise mechanisms by which HATs regulate gene expression and chromatin dynamics in response to light and other environmental signals is essential for developing effective interventions. This knowledge can be applied to enhance crop performance, improve stress tolerance and ensure food security in an increasingly variable climate. Continued interdisciplinary research and collaboration will be crucial in unlocking the full potential of HATs for sustainable agriculture and plant biotechnology.

## Figures and Tables

**Figure 1 ijms-25-07851-f001:**
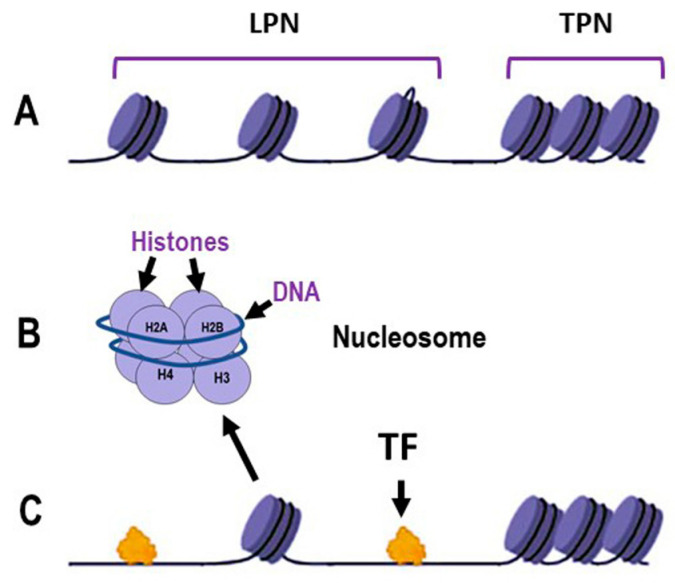
Schematic representation of the dynamics of DNA packaging in nucleosomes. (**A**): Chromatin organization illustrating loosely packed nucleosomes (LPN) and tightly packed nucleosomes (TPN); (**B**): Detailed structure of an individual nucleosome, showing DNA wrapped around a histone octamer composed of histones H2A, H2B, H3 and H4; (**C**): Interaction of transcription factors (TF) with chromatin, demonstrating the accessibility of LPN regions to TF binding, in contrast to the TPN regions. Some elements in this figure have been adapted from BioRender templates (biorender.com).

**Figure 2 ijms-25-07851-f002:**
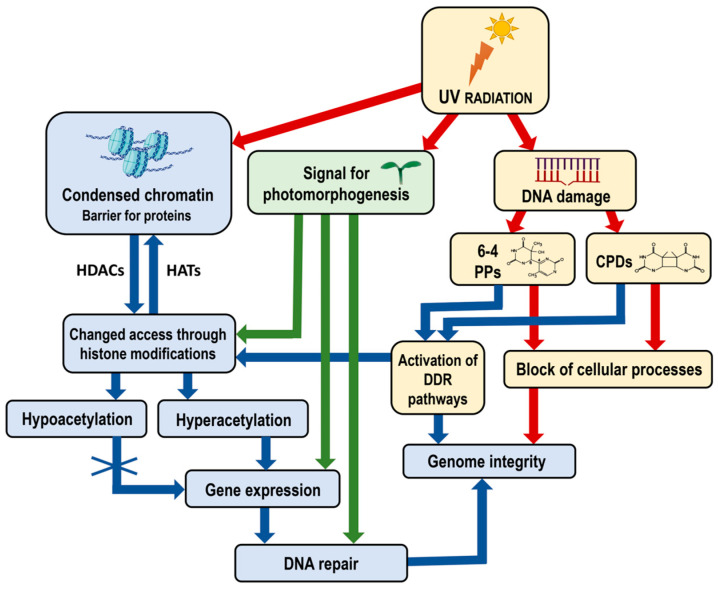
Diagram illustrating the impact of UV radiation on chromatin structure, gene expression and genome integrity, including the specific role of photomorphogenesis signals. UV radiation induces DNA damage, forming 6-4 PPs and CPDs, which block cellular processes and threaten genome integrity. Chromatin, structured into nucleosomes, acts as a barrier to the access of transcription factors and other regulatory proteins, which can be modified by histone modifications. Hyperacetylation allows gene expression necessary for DNA repair, whereas hypoacetylation condenses chromatin, restricting gene expression. UV radiation also signals photomorphogenesis, affecting gene expression related to developmental processes and enhancing repair mechanisms under light conditions. The DDR pathways are activated to recognize and initiate the repair of DNA damage and maintain genome integrity. The interplay between chromatin modifications, UV-induced DNA damage and cellular responses maintains genome integrity and regulates plant development. Red arrows indicate negative effects—blocking cellular processes and compromising genome integrity. Blue and green arrows indicate positive effects—the activation of DDR pathways and chromatin remodeling processes, enhanced gene expression and DNA repair. Abbreviations: UV, Ultraviolet radiation; CPDs, cyclobutane pyrimidine dimers; 6-4 PPs, pyrimidine–pyrimidone (6-4) photoproducts; DDR, DNA damage response; HATs, histone acetyltransferases; HDACs, histone deacetylases. Some elements in this figure have been adapted from BioRender templates (biorender.com).

**Figure 3 ijms-25-07851-f003:**
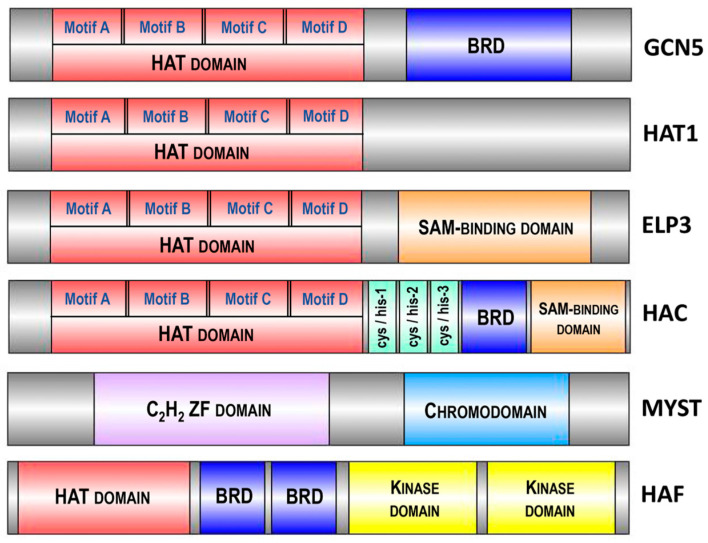
Domain architecture of various histone acetyltransferases (HATs) highlighting conserved motifs and domains. The diagram depicts the domain organization of GCN5, HAC, ELP3, HAT1, MYST and HAF, illustrating the presence of the HAT domain responsible for acetyltransferase activity across different enzymes. Additional domains such as the bromodomain (BRD), SAM-binding domain, C_2_H_2_ zinc finger (ZF) domain, chromodomain and kinase domain are shown, emphasizing the multifunctionality and regulatory mechanisms of these HATs in chromatin modification and gene expression. Conserved motifs (A, B, C, D) in the HAT domain are indicated. The size of different enzymes and domains is schematically presented and does not reflect the actual size of the structural elements of the proteins. Abbreviations: HAT, histone acetyltransferase; BRD, bromodomain; SAM, S-adenosylmethionine; ZF, zinc finger.

**Figure 4 ijms-25-07851-f004:**
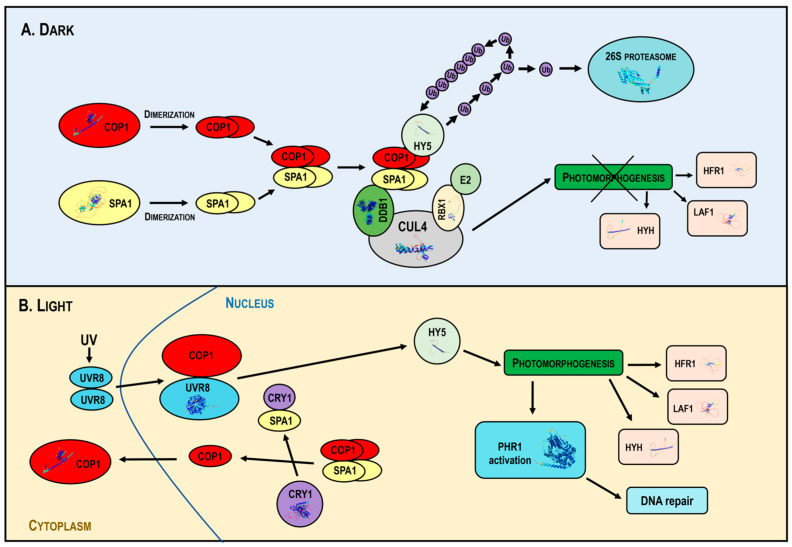
Mechanism of action of the multimeric CUL4-DDB1-RBX1-COP1-SPA1 E3 ligase under dark and light conditions. In the dark (**Panel A**), COP1 forms a complex with SPA proteins and the CUL4-DDB1-RBX1 core, targeting the transcription factors HY5, HYH, LAF1 and HFR1 for ubiquitin-mediated degradation, inhibiting photomorphogenesis. In light (**Panel B**), photoreceptors inhibit COP1, disassembling the COP1-SPA complex and promoting the nuclear export of COP1, allowing the accumulation of photomorphogenic regulators and activating DNA repair mechanisms. Abbreviations: CUL4, Cullin4; DDB1, DNA Damage-Binding Protein 1; RBX1, Ring-Box Protein 1; COP1, Constitutive Photomorphogenic 1; SPA, Suppressor of PhyA-105; HY5, Elongated Hypocotyl 5; Ub, ubiquitin; E2, Ub-conjugating enzyme; HYH, HY5 Homolog; LAF1, Long After Far-Red Light1; HFR1, Long Hypocotyl in Far Red; CRY1, cryptochrome; UVR8, UV RESISTANCE LOCUS 8; PHR1, Photolyase1; UV, Ultraviolet radiation.

**Figure 5 ijms-25-07851-f005:**
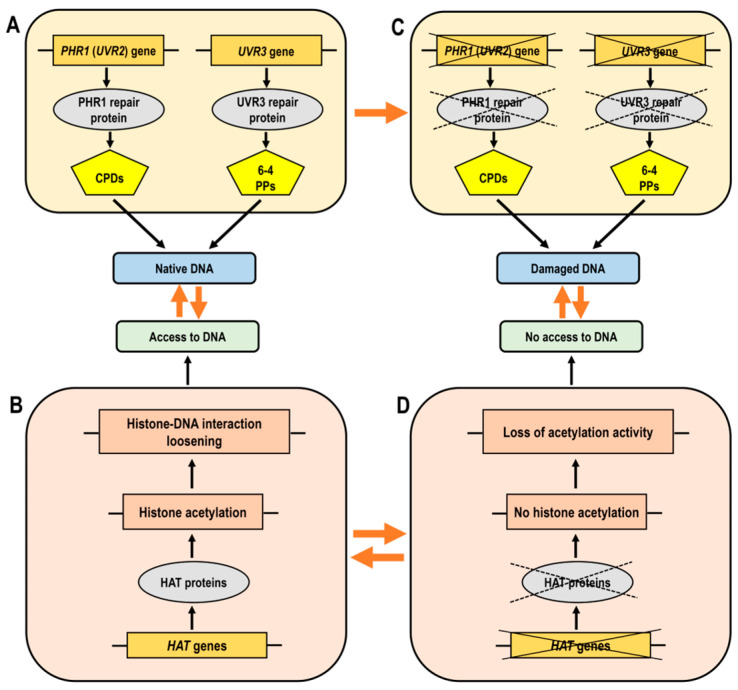
Simplified diagram illustrating the functionality of histone acetyltransferases (HATs) and DNA repair genes, which are essential for effective DNA damage responses. (**Panel A**): Normal chromatin structure where histone acetylation facilitates the loosening of DNA-histone bonds, allowing access of transcription factors to DNA. (**Panel B**): The accessibility of DNA in its native state, supported by proper gene function. (**Panel C**): The effects of DNA damage where mutant forms show a lack of correct response due to impaired HAT function, leading to improper DNA repair. (**Panel D**): Loss or decrease of histone acetylation leads to compacted chromatin and obstructed access of transcription factors to DNA. Abbreviations: HAT, histone acetyltransferase; CPDs, cyclobutane pyrimidine dimers; 6-4 PPs, pyrimidine–pyrimidone (6-4) photoproducts.

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
