# Peer review of "How Histone Acetyltransferases Shape Plant Photomorphogenesis and UV Response"

_ijms, 2024, doi:10.3390/ijms25147851_

Round 1

Reviewer 1 Report

Comments and Suggestions for Authors

This article is a well-written review that explores the roles of HATs in photomorphogenesis, chromatin remodeling, and gene regulation, highlighting the importance of chromatin modifications in plant responses to light and various stressors. It emphasizes the need for further research on individual HAT family members and their interactions with other epigenetic factors. Advanced genomic approaches and genome-editing technologies offer promising avenues for enhancing crop resilience and productivity through targeted manipulation of HAT activities. Understanding these mechanisms is crucial for developing strategies to improve plant growth and stress tolerance, thereby promoting sustainable agriculture in the context of climate change. Here are a few simple suggestions:

1.       Please use vector graphics for all images to ensure greater clarity.

2.       The research objectives in the article appear overly ambitious. The statement in lines 64-66, "we aim to provide a comprehensive understanding of how plants integrate environmental signals to maintain growth, development and genome stability," needs to be more specific.

3.       In Figure 1, the module "Signal for photomorphogenesis" needs to have more specific connections with other modules, and additional details should be provided.

4.       The purpose of the study in lines 518-519, "Further exploration of the diverse roles and mechanisms of HATs can uncover novel strategies to improve plant growth, development and stress resilience, ultimately contributing to sustainable agricultural practices in the face of global environmental challenges," is too broad. It should be revised to relate more directly to the topics discussed in the review, such as photomorphogenesis and UV radiation.

Author Response

Reviewer 1: Comments and responses

Thank you for taking the time to review our manuscript titled ‘How histone acetyltransferases shape plant photomorphogenesis and UV response’ (Manuscript ID: ijms-3115292). We appreciate your constructive comments and have made revisions to address the concerns raised. Below are our responses to each of your points.

Reviewer comment: This article is a well-written review that explores the roles of HATs in photomorphogenesis, chromatin remodeling, and gene regulation, highlighting the importance of chromatin modifications in plant responses to light and various stressors. It emphasizes the need for further research on individual HAT family members and their interactions with other epigenetic factors. Advanced genomic approaches and genome-editing technologies offer promising avenues for enhancing crop resilience and productivity through targeted manipulation of HAT activities. Understanding these mechanisms is crucial for developing strategies to improve plant growth and stress tolerance, thereby promoting sustainable agriculture in the context of climate change. Here are a few simple suggestions:

Response: We sincerely thank the Reviewer for the positive feedback and have carefully considered the provided recommendations.

Reviewer comment 1: Please use vector graphics for all images to ensure greater clarity.

Response 1: We have converted all images to vector graphics to ensure greater clarity in the figures.

Reviewer comment 2: The research objectives in the article appear overly ambitious. The statement in lines 64-66, ‘we aim to provide a comprehensive understanding of how plants integrate environmental signals to maintain growth, development and genome stability,’ needs to be more specific.

Response 2: We appreciate your feedback regarding the specificity of our research objectives. To address this concern, we have refined our objectives to clearly outline the specific aspects of the manuscript. The revised text reads: ‘…we aim to elucidate the specific roles of HATs in modulating chromatin dynamics and gene expression, focusing on their involvement in light signaling pathways, photomorphogenesis and UV-induced DNA repair mechanisms in higher plants’ (lines 75-78).

Reviewer comment 3: In Figure 1, the module ‘Signal for photomorphogenesis’ needs to have more specific connections with other modules, and additional details should be provided.

Response 3: We have completely revised Figure 1 (now Figure 2) to address the Reviewer's concerns. The ‘Signal for photomorphogenesis’ module is now connected to other modules, illustrating the board effects of light signals. Specifically, it is linked to ‘Gene expression’ to show that light signals affect gene regulatory mechanisms, and to ‘DNA repair’ to indicate enhanced repair mechanisms under light conditions. Additionally, the figure depicts the role of ‘Signal for photomorphogenesis’ in modifying chromatin accessibility through histone modifications, affecting the access of transcription factors and other regulatory proteins. We have also included an additional module - the DDR pathway, which is activated to address the DNA damage and contribute to maintaining genome integrity. Thus, the figure now includes more connections among all modules to better illustrate the integrated nature of these processes.

Finally, the figure legend has been updated to reflect all these modifications (lines 187-203). We hope these changes provide clearer connections and additional details as requested.

Reviewer comment 4: The purpose of the study in lines 518-519, ‘Further exploration of the diverse roles and mechanisms of HATs can uncover novel strategies to improve plant growth, development and stress resilience, ultimately contributing to sustainable agricultural practices in the face of global environmental challenges’, is too broad. It should be revised to relate more directly to the topics discussed in the review, such as photomorphogenesis and UV radiation.

Response 4: We have now revised the statement to reflect this remark as follows: ‘Further exploration of the diverse roles and mechanisms of HATs, particularly their influence on photomorphogenesis and UV-induced DNA repair, can uncover novel strategies to enhance plant growth and development, and improve stress resilience, contributing to sustainable agricultural practices in the face of global environmental challenges’ (lines 995-999).

Reviewer 2 Report

Comments and Suggestions for Authors

The manuscript titled “How Histone Acetyltransferases Shape Plant Photomorphogenesis and UV Response” by Boycheva, I.; et al. is a Review work work the authors discussed the most recent advances in the strategies followed by plant systems for DNA repairment when the tissues are exposed to damage ultraviolet radiation. This is a topic of growing interest. However, it exists some points that need to be addressed (please, see them below detailed point-by-point) to improve the scientific quality of the submitted manuscript paper before this article will be consider for its publication in the International Journal of Molecular Sciences.

1) The authors should consider to modify the term “photomorphogenesis” by “plant photomorphogenesis” in the keyword list.

2) “To counteract these effects (…) DNA repair mechanisms (…) regulation of gene expression and plant stress adaptability” (lines 37-41). Here, even if I agree with this statement provided by the authors it should not be neglected the contribution of histones (specially the class H2A), not only in DNA reparation [1], but also in the degradosome formation with the subsequent chromatinolysis process [2].

[1] Selvam, K.; et al. DNA Repair in Nucleosomes : Insights from Histone Modifications and Mutants. Int. J. Mol. Sci. 2024, 25, 4393. https://doi.org/10.3390/ijms25084393.

[2] Novo, N.; et al. Beyond a platform protein for the degradosome assembly: The Apoptosis-Inducing Factor as an efficient nuclease involved in chromatinolysis. PNAS Nexus 2022, 2, pgac312. https://doi.org/10.1093/pnasnexus/pgac312.

3) “2.1. Chromatin structure as a natural barrier to transcription factor access” (lines 68-104). It would be desirable if the authors could furnish with some images (e.g. SEM or any other single molecule technique) about the different degrees of DNA densities by the nucleosomes.

4) “3. HAT families in plants” (lines 141-184). What is the preservation degree of the most conserved motifs in plants? A brief statement should be provided in this regard.

5) “6. HATs and DNA repair capacity” (lines 432-471). A schematic representation will aid the potential readers to better understand the underlying mechanisms explained in this section.

6) “7. Future perspectives and 8. Conclusion” (lines 472-525). These section perfectly remarks the most relevant outcomes found by the authors in this field and the promising future perspectives. It should be desirable to add a brief statement to discuss about the potential future action lines to pursue the topic of this research

Comments on the Quality of English Language

The manuscript is well-written albeit it may be advisable a final check to polish those final details susceptible to be improved.

Author Response

Reviewer 2: Comments and responses

Thank you for the thorough review of our manuscript titled ‘How histone acetyltransferases shape plant photomorphogenesis and UV response’ (Manuscript ID: ijms-3115292). We value the insightful comments and have made revisions to address the issues raised. Below are our detailed responses to each of the points.

Reviewer comment: The manuscript titled ‘How Histone Acetyltransferases Shape Plant Photomorphogenesis and UV Response’ by Boycheva, I.; et al. is a Review work work the authors discussed the most recent advances in the strategies followed by plant systems for DNA repairment when the tissues are exposed to damage ultraviolet radiation. This is a topic of growing interest. However, it exists some points that need to be addressed (please, see them below detailed point-by-point) to improve the scientific quality of the submitted manuscript paper before this article will be consider for its publication in the International Journal of Molecular Sciences.

Response: Thank you for your constructive feedback and for recognising the importance of our review manuscript. We have carefully considered your detailed comments and have addressed each point to improve the quality of our manuscript. Please find our responses and the corresponding revisions below.

Reviewer comment 1: 1) The authors should consider to modify the term ‘photomorphogenesis’ by ‘plant photomorphogenesis’ in the keyword list.

Response 1: We agree that specifying ‘plant photomorphogenesis’ would enhance the precision of our keywords. This modification has been made (line 25).

Reviewer comment 2: 2) ‘To counteract these effects (…) DNA repair mechanisms (…) regulation of gene expression and plant stress adaptability’ (lines 37-41). Here, even if I agree with this statement provided by the authors it should not be neglected the contribution of histones (specially the class H2A), not only in DNA reparation [1], but also in the degradosome formation with the subsequent chromatinolysis process [2].

[1] Selvam, K.; et al. DNA Repair in Nucleosomes : Insights from Histone Modifications and Mutants. Int. J. Mol. Sci. 2024, 25, 4393. https://doi.org/10.3390/ijms25084393.

[2] Novo, N.; et al. Beyond a platform protein for the degradosome assembly: The Apoptosis-Inducing Factor as an efficient nuclease involved in chromatinolysis. PNAS Nexus 2022, 2, pgac312.’

Response 2: Thank you for your comment and for highlighting the important contributions of histones in degradosome formation and chromatinolysis. We have revised the text to include these aspects, ensuring a more comprehensive discussion. The following changes have been made to the manuscript: ‘To counteract these effects, plants have developed specific DNA repair mechanisms that protect them from this damage and ensure normal growth and development [10-13]. The repair mechanisms are closely related to chromatin remodeling events (e.g., DNA methylation and histone acetylation), which contribute to the regulation of gene expression and plant stress adaptability. It is also important to acknowledge the role of histones, particularly the H2A class, not only in DNA repair processes [14] but also in degradosome formation and subsequent chromatinolysis [15]. The degradosome is es-sential for chromatin degradation during programmed cell death. In this context, the Apoptosis-Inducing Factor (AIF) collaborates with histone H2AX and cyclophilin A (CypA) to form a DNA–degradosome complex, which is vital for large-scale DNA fragmentation. This complex enhances the affinity of degradosome for DNA, system-atically degrading damaged DNA and associated histones, thus preventing the accu-mulation of damaged chromatin, and promoting cellular recovery and survival [15].’

Reviewer comment 3: 3) ’2.1. Chromatin structure as a natural barrier to transcription factor access’ (lines 68-104). It would be desirable if the authors could furnish with some images (e.g. SEM or any other single molecule technique) about the different degrees of DNA densities by the nucleosomes.

Response 3: To address this comment, we have incorporated historical electron micrographs that demonstrate the varying degrees of DNA packing within nucleosomes, as reported by Olins and Olins [23]. These images provide a visual representation of chromatin structure and its potential impact on DNA accessibility, which we believe will enhance the understanding of chromatin dynamics as a natural barrier to transcription factor access. These images can be found in Figure 1 of the revised manuscript. We have also provided relevant text in lines 84-91.

Reviewer comment 4: 4) ‘3. HAT families in plants’ (lines 141-184). What is the preservation degree of the most conserved motifs in plants? A brief statement should be provided in this regard.

Response 4: We have added a brief statement to section 3 to address this aspect (lines 277-282 & 324-326). This addition provides the necessary context regarding the conservation of HAT motifs in plants. We believe this enhancement improves the depth of our review and ensures that the reader understands the importance and prevalence of conserved motifs across various plant species.

Reviewer comment 5:  5) ‘6. HATs and DNA repair capacity’ (lines 432-471). A schematic representation will aid the potential readers to better understand the underlying mechanisms explained in this section.

Response 5: To enhance reader understanding of the mechanisms discussed in this section, we have added Figure 5, which schematically illustrates the role of HATs and DNA repair genes in DNA damage responses. The figure provides a visual representation of normal chromatin structure, the effects of histone acetylation on DNA accessibility and the impact of impaired HAT function on DNA repair processes. This addition should help readers better grasp the complex interactions involved in HAT-mediated DNA repair.

Reviewer comment 6: 6) ‘7. Future perspectives and 8. Conclusion’ (lines 472-525). These section perfectly remarks the most relevant outcomes found by the authors in this field and the promising future perspectives. It should be desirable to add a brief statement to discuss about the potential future action lines to pursue the topic of this research

Response 6: We have slightly revised the ‘7. Future perspectives’ (lines 988-993) and ‘8. Conclusion’ (lines 997-999) sections.

In the Future perspectives section, we have expanded the text to outline specific research lines, which include:

  • Development of systematic models incorporating various epigenetic layers and modifications.
  • Further study of the interplay between chromatin dynamics and DNA repair for UV tolerance.
  • Detailed characterization of HAT family members across different plant species.
  • Investigation of the interaction between HATs and other epigenetic factors.
  • Utilization of advanced genomic and proteomic approaches like ChIP-seq and mass spectrometry.
  • Modulation of DNA repair and stress-responsive genes, including developing HAT inhibitors/activators.
  • Application of genome-editing technologies such as CRISPR/Cas9.
  • Integration of multi-omics approaches to understand HAT functions.
  • Collaboration with agricultural scientists to translate research into practical applications.

In the Conclusion section, we emphasised the value of understanding the precise mechanisms by which HATs regulate gene expression and chromatin dynamics in response to light and other environmental signals. We reiterated the importance of interdisciplinary research and collaboration to unlock the full potential of HATs for sustainable agriculture and plant biotechnology.

We hope that these additions align with Reviewer expectations and provide a roadmap for future research in this field.

Reviewer: Comments on the quality of English language

The manuscript is well-written albeit it may be advisable a final check to polish those final details susceptible to be improved.

Response: We have carefully reviewed the manuscript once more to ensure clarity and precision in the language. We have made the necessary adjustments to improve readability and overall quality.